# Near-Optimal Smoothing of Structured Conditional Probability Matrices

**Moein Falahatgar**
University of California, San Diego
San Diego, CA, USA
moein@ucsd.edu

**Mesrob I. Ohannessian**
Toyota Technological Institute at Chicago
Chicago, IL, USA
mesrob@ttic.edu

**Alon Orlitsky**
University of California, San Diego
San Diego, CA, USA
alon@ucsd.edu

## Abstract

Utilizing the structure of a probabilistic model can significantly increase its learning speed. Motivated by several recent applications, in particular bigram models in language processing, we consider learning low-rank conditional probability matrices under expected KL-risk. This choice makes smoothing, that is the careful handling of low-probability elements, paramount. We derive an iterative algorithm that extends classical non-negative matrix factorization to naturally incorporate additive smoothing and prove that it converges to the stationary points of a penalized empirical risk. We then derive sample-complexity bounds for the global minimzer of the penalized risk and show that it is within a small factor of the optimal sample complexity. This framework generalizes to more sophisticated smoothing techniques, including absolute-discounting.

## 1 Introduction

One of the fundamental tasks in statistical learning is probability estimation. When the possible outcomes can be divided into $k$ discrete categories, e.g. types of words or bacterial species, the task of interest is to use data to estimate the probability masses $p_1, \cdots, p_k$, where $p_j$ is the probability of observing category $j$. More often than not, it is not a single distribution that is to be estimated, but multiple *related* distributions, e.g. frequencies of words within various contexts or species in different samples. We can group these into a conditional probability (row-stochastic) matrix $P_{i,1}, \cdots, P_{i,k}$ as $i$ varies over $c$ contexts, and $P_{ij}$ represents the probability of observing category $j$ in context $i$. Learning these distributions individually would cause the data to be unnecessarily diluted. Instead, the *structure* of the relationship between the contexts should be harnessed.

A number of models have been proposed to address this structured learning task. One of the wildly successful approaches consists of positing that $P$, despite being a $c \times k$ matrix, is in fact of much lower rank $m$. Effectively, this means that there exists a latent context space of size $m \ll c, k$ into which the original context maps probabilistically via a $c \times m$ stochastic matrix $A$, then this latent context in turn determines the outcome via an $m \times k$ stochastic matrix $B$. Since this structural model means that $P$ factorizes as $P = AB$, this problem falls within the framework of low-rank (non-negative) matrix factorization. Many topic models, such as the original work on probabilistic latent semantic analysis PLSA, also map to this framework. We narrow our attention here to such low-rank models, but note that more generally these efforts fall under the areas of structured and transfer learning. Other examples include: manifold learning, multi-task learning, and hierarchical models.

In natural language modeling, low-rank models are motivated by the inherent semantics of language: context first maps into meaning which then maps to a new word prediction. An alternative form of such latent structure, word embeddings derived from recurrent neural networks (or LSTMs) are the state-of-the-art of current language models, [20, 25, 28]. A first chief motivation for the present work is to establish a theoretical underpinning of the success of such representations. We restrict the exposition to *bigram* models. The traditional definition of the bigram is that language is modeled as a sequence of words generated by a first order Markov-chain. Therefore the 'context' of a new word is simply its preceding word, and we have $c = k$. Since the focus here is not the dependencies induced by such memory, but rather the ramifications of the structural assumptions on $P$, we take bigrams to model word-pairs *independently* sampled by first choosing the contextual word with probability $\pi$ and then choosing the second word according to the conditional probability $P$, thus resulting in a joint distribution over word-pairs $(\pi_i P_{ij})$.

What is the natural measure of performance for a probability matrix estimator? Since ultimately such estimators are used to accurately characterize the likelihood of test data, the measure of choice used in empirical studies is the perplexity, or alternatively its logarithm, the cross entropy. For data consisting of $n$ word-pairs, if $C_{ij}$ is the number of times pair $(i, j)$ appears, then the cross entropy of an estimator $Q$ is $\frac{1}{n} \sum_{ij} C_{ij} \log \frac{1}{Q_{ij}}$. The population quantity that corresponds to this empirical performance measure is the (row-by-row weighted) KL-divergence $D(P\|Q) = \sum_{ij} \pi_i P_{ij} \log \frac{P_{ij}}{Q_{ij}}$.

Note that this is indeed the expectation of the cross entropy modulo the true entropy, an additive term that does not depend on $Q$. This is the natural notion of *risk* for the learning task, since we wish to infer the likelihood of future data, and our goal can now be more concretely stated as using the data to produce an estimator $Q_n$ with a 'small' value of $D(P\|Q_n)$. The choice of KL-divergence introduces a peculiar but important problem: the necessity to handle small frequencies appropriately. In particular, using the empirical conditional probability is not viable, since a zero in $Q$ implies infinite risk. This is the problem of *smoothing*, which has received a great amount of attention by the NLP community. Our second salient motivation for the present work is to propose principled methods of integrating well-established smoothing techniques, such as *add-$\frac{1}{2}$* and *absolute discounting*, into the framework of structured probability matrix estimation.

Our contributions are as follows, we provide:

- A general framework for integrating smoothing and structured probability matrix estimation, as an alternating-minimization that converges to a stationary point of a penalized empirical risk.

- A sample complexity upper bound of $\mathcal{O}(km \log^2(2n + k)/n)$ for the expected KL-risk, for the global minimizer of this penalized empirical risk.

- A lower bound that matches this upper bound up to the logarithmic term, showing near-optimality.

The paper is organized as follows. Section 2 reviews related work. Section 3 states the problem and Section 4 highlights our main results. Section 5 proposes our central algorithm and Section 6 analyzes its idealized variant. Section 7 provides some experiments and Section 8 concludes.

## 2 Related Work

Latent variable models, and in particular non-negative matrix factorization and topic models, have been such an active area of research in the past two decades that the space here cannot possibly do justice to the many remarkable contributions. We list here some of the most relevant to place our work in context. We start by mentioning the seminal papers [12, 18] which proposed the alternating minimization algorithm that forms the basis of the current work. This has appeared in many forms in the literature, including the multiplicative updates [29]. Some of the earliest work is reviewed in [23]. These may be generally interpreted as discrete analogs to PCA (and even ICA) [10].

An influential Bayesian generative topic model, the Latent Dirichlet Allocation, [7] is very closely related to what we propose. In fact, add-half smoothing effectively corresponds to a Dirichlet(1/2) (Jeffreys) prior. Our exposition differs primarily in adopting a minimax sample complexity perspective which is often not found in the otherwise elegant Bayesian framework. Furthermore, exact Bayesian inference remains a challenge and a lot of effort has been expended lately toward simple iterative algorithms with provable guarantees, e.g. [3, 4]. Besides, a rich array of efficient smoothing techniques exists for probability vector estimation [2, 16, 22, 26], of which one could directly avail in the methodology that is presented here.

A direction that is very related to ours was recently proposed in [13]. There, the primary goal is to recover the rows of $A$ and $B$ in $\ell_1$-risk. This is done at the expense of additional separation conditions on these rows. This makes the performance measure not easily comparable to our context, though with the proper weighted combination it is easy to see that the implied $\ell_1$-risk result on $P$ is subsumed by our KL-risk result (via Pinsker's inequality), up to logarithmic factors, while the reverse isn't true. Furthermore, the framework of [13] is restricted to symmetric joint probability matrices, and uses an SVD-based algorithm that is difficult to scale beyond very small latent ranks $m$. Apart from this recent paper for the $\ell_1$-risk, sample complexity bounds for related (not fully latent) models have been proposed for the KL-risk, e.g. [1]. But these remain partial, and far from optimal. It is also worth noting that information geometry gives conditions under which KL-risk behaves close to $\ell_2$-risk [8], thus leading to a Frobenius-type risk in the matrix case.

Although the core optimization problem itself is not our focus, we note that despite being a non-convex problem, many instances of matrix factorization admit efficient solutions. Our own heuristic initialization method is evidence of this. Recent work, in the $\ell_2$ context, shows that even simple gradient descent, appropriately initialized, could often provably converge to the global optimum [6].

Concerning whether such low-rank models are appropriate for language modeling, there has been evidence that some of the abovementioned word embeddings [20] can be interpreted as implicit matrix factorization [19]. Some of the traditional bigram smoothing techniques, such as the Kneser-Ney algorithm [17, 11], are also reminiscent of rank reduction [14, 24, 15].

## 3  Problem Statement

Data $\mathcal{D}_n$ consists of $n$ pairs $(X_s, Y_s)$, $s = 1, \cdots, n$, where $X_s$ is a context and $Y_s$ is the corresponding outcome. In the spirit of a bigram language model, we assume that the context and outcome spaces have the same cardinality, namely $k$. Thus $(X_s, Y_s)$ takes values in $[k]^2$. We denote the count of pairs $(i, j)$ by $C_{ij}$. As a shortcut, we also write the row-sums as $C_i = \sum_j C_{ij}$.

We assume the underlying generative model of the data to be i.i.d., where each pair is drawn by first sampling the context $X_s$ according to a probability distribution $\pi = (\pi_i)$ over $[k]$ and then sampling $Y_s$ conditionally on $X_s$ according to a $k \times k$ conditional probability (stochastic) matrix $P = (P_{ij})$, a non-negative matrix where each row sums to 1. We also assume that $P$ has non-negative rank $m$. We denote the set of all such matrices by $\mathcal{P}_m$. They can all be factorized (non-uniquely) as $P = AB$, where both $A$ and $B$ are stochastic matrices in turn, of size $k \times m$ and $m \times k$ respectively.

A conditional probability matrix estimator is an algorithm that maps the data into a stochastic matrix $Q_n(X_1, \cdots, X_n)$ that well-approximates $P$, in the absence of any knowledge about the underlying model. We generally drop the explicit notation showing dependence on the data, and use instead the implicit $n$-subscript notation. The performance, or how well any given stochastic matrix $Q$ approximates $P$, is measured according to the KL-risk:

$$R(Q) = \sum_{ij} \pi_i P_{ij} \log \frac{P_{ij}}{Q_{ij}} \tag{1}$$

Note that this corresponds to an expected loss, with the log-loss $L(Q, i, j) = \log P_{ij}/Q_{ij}$. Although we do seek out PAC-style (in-probability) bounds for $R(Q_n)$, in order to give a concise definition of optimality, we consider the average-case performance $\mathbb{E}[R(Q_n)]$. The expectation here is with respect to the data. Since the underlying model is completely unknown, we would like to do well against adversarial choices of $\pi$ and $P$, and thus we are interested in a uniform upper bound of the form:

$$r(Q_n) = \max_{\pi, P \in \mathcal{P}_m} \mathbb{E}[R(Q_n)].$$

The optimal estimator, in the minimax sense, and the minimax risk of the class $\mathcal{P}_m$ are thus given by:

$$Q_n^\star = \arg\min_{Q_n} r(Q_n) \quad = \quad \arg\min_{Q_n} \max_{\pi, P \in \mathcal{P}_m} \mathbb{E}[R(Q_n)]$$
$$r^\star(\mathcal{P}_m) \quad = \quad \min_{Q_n} \max_{\pi, P \in \mathcal{P}_m} \mathbb{E}[R(Q_n)].$$

Explicitly obtaining minimax optimal estimators is a daunting task, and instead we would like to exhibit estimators that compare well.

**Definition 1** (Optimality). *If an estimator satisfies $\mathbb{E}[R(Q_n)] \leq \varphi \cdot \mathbb{E}[R(Q_n^\star)]$, $\forall \pi$, (called an* oracle *inequality), then if $\varphi$ is a constant (of $n$, $k$, and $m$), we say that the estimator is (order)* optimal. *If $\varphi$ is not constant, but its growth is negligible with respect to the decay of $r^\star(\mathcal{P}_m)$ with $n$ or the growth of $r^\star(\mathcal{P}_m)$ with $k$ or $m$, then we can call the estimator* near-optimal. *In particular, we reserve this terminology for a logarithmic gap in growth, that is an estimator is near-optimal if $\log \varphi / \log r^\star(\mathcal{P}_m) \to 0$ asymptotically in any of $n$, $k$, or $m$. Finally, if $\varphi$ does not depend on $P$ we have* strong *optimality, and $r(Q_n) \leq \varphi \cdot r^\star(\mathcal{P}_m)$. If $\varphi$ does depend on $P$, we have* weak *optimality.*

As a proxy to the true risk (1), we define the empirical risk:

$$R_n(Q) = \frac{1}{n} \sum_{ij} C_{ij} \log \frac{P_{ij}}{Q_{ij}} \tag{2}$$

The conditional probability matrix that minimizes this empirical risk is the empirical conditional probability $\hat{P}_{n,ij} = C_{ij}/C_i$. Not only is $\hat{P}_{n,ij}$ not optimal, but since there always is a positive (even if slim) probability that some $C_{ij} = 0$ even if $P_{ij} \neq 0$, it follows that $\mathbb{E}[R_n(\hat{P}_n)] = \infty$. This shows the importance of smoothing. The simplest benchmark smoothing that we consider is *add-$\frac{1}{2}$* smoothing $\hat{P}_{ij}^{\mathsf{Add}\text{-}\frac{1}{2}} = (C_{ij} + 1/2)\,/\,(C_i + k/2)$, where we give an additional "phantom" half-sample to each word-pair, to avoid zeros. This simple method has optimal minimax performance when estimating probability vectors. However, in the present matrix case it is possible to show that this can be a factor of $k/m$ away from optimal, which is significant (cf. Figure 1(a) in Section 7). Of course, since we have not used the low-rank structure of $P$, we may be tempted to "smooth by factoring", by performing a low-rank approximation of $\hat{P}_n$. However, this will not eliminate the zero problem, since a whole column may be zero. These facts highlight the importance of principled smoothing. The problem is therefore to construct (possibly weakly) optimal or near-optimal smoothed estimators.

## 4 Main Results

In Section 5 we introduce the ADD-$\frac{1}{2}$-SMOOTHED LOW-RANK algorithm, which essentially consists of EM-style alternating minimizations, with the addition of smoothing at each stage. Here we state the main results. The first is a characterization of the implicit risk function that the algorithm targets.

**Theorem 2** (Algorithm). *$Q^{\mathsf{Add}\text{-}\frac{1}{2}\text{-}\mathsf{LR}}$ converges to a stationary point of the* penalized *empirical risk*

$$R_{\mathsf{n,penalized}}(W, H) = R_n(Q) + \frac{1}{2n} \sum_{i,\ell} \log \frac{1}{W_{i\ell}} + \frac{1}{2n} \sum_{\ell,j} \log \frac{1}{H_{\ell j}}, \quad \text{where } Q = WH. \tag{3}$$

*Conversely, any stationary point of (3) is a stable point of* ADD-$\frac{1}{2}$-SMOOTHED LOW-RANK.

The proof of Theorem 2 follows closely that of [18]. We now consider the global minimum of this implicit risk, and give a sample complexity bound. By doing so, we intentionally decouple the algorithmic and statistical aspects of the problem and focus on the latter.

**Theorem 3** (Sample Complexity). *Let $Q_n \in \mathcal{P}_m$ achieve the global minimum of Equation 3. Then for all $P \in \mathcal{P}_m$ such that $P_{ij} > \frac{km}{n} \log(2n + k) \; \forall i, j$ and $n > 3$,*

$$\mathbb{E}[R(Q_n)] \leq \bar{c} \frac{km}{n} \log^2(2n + k), \qquad \text{with } \bar{c} = 3100.$$

We outline the proof in Section 6. The basic ingredients are: showing the problem is near-realizable, a quantization argument to describe the complexity of $\mathcal{P}_m$, and a PAC-style [27] relative uniform convergence which uses a sub-Poisson concentration for the sums of log likelihood ratios and uniform variance and scale bounds. Finer analysis based on VC theory may be possible, but it would need to handle the challenge of the log-loss being possibly unbounded and negative. The following result shows that Theorem 3 gives weak near-optimality for $n$ large, as it is tight up to the logarithmic factor.

**Theorem 4** (Lower Bound). *For $n > k$, the minimax rate of $\mathcal{P}_m$ satisfies:*

$$r^\star(\mathcal{P}_m) \geq \underline{c} \frac{km}{n}, \qquad \text{with } \underline{c} = 0.06.$$

This is based on the vector case lower bound and providing the oracle with additional information: instead of only $(X_s, Y_s)$ it observes $(X_s, Z_s, Y_s)$, where $Z_s$ is sampled from $X_s$ using $A$ and $Y_s$ is sampled from $Z_s$ using $B$. This effectively allows the oracle to estimate $A$ and $B$ directly.

# 5 Algorithm

Our main algorithm is a direct modification of the classical alternating minimization algorithm for non-negative matrix factorization [12, 18]. This classical algorithm (with a slight variation) can be shown to essentially solve the following mathematical program:

$$Q^{\mathsf{NNMF}}(\Phi) = \arg\min_{Q=WH} \sum_i \sum_j \Phi_{ij} \log \frac{1}{Q_{ij}}.$$

The analysis is a simple extension of the original analysis of [12, 18]. By "essentially solves", we mean that each of the update steps can be identified as a coordinate descent, reducing the cost function and ultimately converging as $T \to \infty$ to a stationary (zero gradient) point of this function. Conversely, all stationary points of the function are stable points of the algorithm. In particular, since the problem is convex in $W$ and $H$ individually, but not jointly in both, the algorithm can be thought of as taking exact steps toward minimizing over $W$ (as $H$ is held fixed) and then minimizing over $H$ (as $W$ is held fixed), whence the *alternating-minimization* name.

Before we incorporate smoothing, note that there are two ingredients missing from this algorithm. First, the cost function is the sum of row-by-row KL-divergences, but each row is *not weighted*, as compared to Equation (1). If we think of $\Phi_{ij}$ as $\hat{P}_{ij} = C_{ij}/C_i$, then the natural weight of row $i$ is $\pi_i$ or its proxy $C_i/n$. For this, the algorithm can easily be patched. Similarly to the analysis of the original algorithm, one finds that this change essentially minimizes the *weighted* KL-risks of the empirical conditional probability matrix, or equivalently the empirical risk as defined in Equation (2):

$$Q^{\mathsf{LR}}(C) = \arg\min_{Q=WH} R_n(Q) = \arg\min_{Q=WH} \sum_i \frac{C_i}{n} \sum_j \frac{C_{ij}}{C_i} \log \frac{1}{Q_{ij}}.$$

Of course, this is nothing but the maximum likelihood estimator of $P$ under the low-rank constraint. Just like the empirical conditional probability matrix, it suffers from lack of smoothing. For instance, if a whole column of $C$ is zero, then so will be the corresponding column of $Q^{\mathsf{ERM}}(C)$. The first naive attempt at smoothing would be to add-$\frac{1}{2}$ to $C$ and then apply the algorithm:

$$Q^{\mathsf{Naive\ Add\text{-}}\frac{1}{2}\mathsf{-LR}}(C) = Q^{\mathsf{LR}}(C + \tfrac{1}{2})$$

However, this would result in excessive smoothing, especially when $m$ is small. The intuitive reason is this: in the extreme case of $m = 1$ all rows need to be combined, and thus instead of adding $\frac{1}{2}$ to each category, $Q^{\mathsf{Naive\ add}-\frac{1}{2}\mathsf{LR}}$ would add $k/2$, leading to the the uniform distribution overwhelming the original distribution. We may be tempted to mitigate this by adding instead $1/2k$, but this doesn't generalize well to other smoothing methods. A more principled approach should perform smoothing directly *inside* the factorization, and this is exactly what we propose here. Our main algorithm is:

**Algorithm:** ADD-$\frac{1}{2}$-SMOOTHED LOW-RANK

- Input: $k \times k$ matrix $(C_{ij})$; Initial $W^0$ and $H^0$; Number of iterations $T$
- Iterations: Start at $t = 0$, increment and repeat while $t < T$
    - For all $i \in [k], \ell \in [m]$, update $W_{i\ell}^t \leftarrow W_{i\ell}^{t-1} \sum_j \frac{C_{ij}}{(WH)_{ij}^{t-1}} H_{\ell j}^{t-1}$
    - For all $\ell \in [m], j \in [k]$, update $H_{\ell j}^t \leftarrow H_{\ell j}^{t-1} \sum_i \frac{C_{ij}}{(WH)_{ij}^{t-1}} W_{i\ell}^{t-1}$
    - Add-1/2 to each element of $W^t$ and $H^t$, then normalize each row.
- Output: $Q^{\mathsf{Add\text{-}}\frac{1}{2}\mathsf{-LR}}(C) = W^T H^T$

The intuition here is that, prior to normalization, the updated $W$ and $H$ can be interpreted as *soft counts*. One way to see this is to sum each row $i$ of (pre-normalized) $W$, which would give $C_i$. As for $H$, the sums of its (pre-normalized) columns reproduce the sums of the columns of $C$. Next, we are naturally led to ask: is $Q^{\mathsf{Add\text{-}}\frac{1}{2}\mathsf{LR}}(C)$ implicitly minimizing a risk, just as $Q^{\mathsf{LR}}(C)$ minimizes $R_n(Q)$? Theorem 2 shows that indeed $Q^{\mathsf{Add\text{-}}\frac{1}{2}\mathsf{LR}}(C)$ essentially minimizes a *penalized* empirical risk.

More interestingly, ADD-$\frac{1}{2}$-SMOOTHED LOW-RANK lends itself to a host of generalizations. In particular, an important smoothing technique, *absolute discounting*, is very well suited for heavy-tailed data such as natural language [11, 21, 5]. We can generalize it to fractional counts as follows. Let $C_i$ indicate counts in traditional (vector) probability estimation, and let $D$ be the total number of

distinct observed categories, i.e. $D = \sum_i \mathbb{I}\{C_i \geq 1\}$. Let the number of *fractional* distinct categories d be defined as $\mathrm{d} = \sum_i C_i \mathbb{I}\{C_i < 1\}$. We have the following *soft absolute discounting* smoothing:

$$\hat{P}_i^{\mathsf{Soft\text{-}AD}}(C, \alpha) = \begin{cases} \frac{C_i - \alpha}{\sum C} & \text{if } C_i \geq 1, \\ \frac{1-\alpha}{\sum C} C_i + \frac{\alpha(D+\mathrm{d})}{(k - D - \mathrm{d})\sum C}(1 - C_i) & \text{if } C_i < 1. \end{cases}$$

This gives us the following patched algorithm, which we do not place under the lens of theory currently, but we strongly support it with our experimental results of Section 7.

**Algorithm:** ABSOLUTE-DISCOUNTING-SMOOTHED LOW-RANK

- Input: Specify $\alpha \in (0, 1)$
- Iteration:
    - Add-$1/2$ to each element of $W^t$, then normalize.
    - Apply soft absolute discounting to $H_{\ell j}^t \leftarrow \hat{P}_j^{\mathsf{Soft\text{-}AD}}(H_{\ell,\cdot}^t, \alpha)$
- Output: $Q^{\mathsf{AD\text{-}LR}}(C, \alpha) = W^T H^T$

## 6 Analysis

We now outline the proof of the sample complexity upper bound of Theorem 3. Thus for the remainder of this section we have:

$$Q_n(C) = \underset{Q = WH}{\arg\min}\, R_n(Q) + \frac{1}{2n}\sum_{i,\ell}\log\frac{1}{W_{i\ell}} + \frac{1}{2n}\sum_{\ell,j}\log\frac{1}{H_{\ell j}},$$

that is $Q_n \in \mathcal{P}_m$ achieves the global minimum of Equation 3. Since we have a penalized empirical risk minimization at hand, we can study it within the classical PAC-learning framework. However, rates of order $\frac{1}{n}$ are often associated withe the *realizable* case, where $R_n(Q_n)$ is exactly zero [27]. The following Lemma shows that we are *near* the realizable regime.

**Lemma 5** (Near-realizability). *We have*

$$\mathbb{E}[R_n(Q_n)] \leq \frac{k}{n} + \frac{km}{n}\log(2n + k).$$

We characterize the complexity of the class $\mathcal{P}_m$ by *quantizing* probabilities, as follows. Given a positive integer $L$, define $\Delta_L$ to be the subset of the appropriate simplex $\Delta$ consisting of $L$-empirical distributions (or "types" in information theory): $\Delta_L$ consists exactly of those distributions $p$ that can be written as $p_i = L_i/L$, where $L_i$ are non-negative integers that sum to $L$.

**Definition 6** (Quantization). *Given a positive integer $L$, define the $L$-quantization operation as mapping a probability vector $p$ to the closest (in $\ell_1$-distance) element of $\Delta_L$, $\tilde{p} = \arg\min_{q \in \Delta_L}\|p - q\|_1$. For a matrix $P \in \mathcal{P}_m$, define an $L$-quantization for any given factorization choice $P = AB$ as $\tilde{P} = \tilde{A}\tilde{B}$, where each row of $\tilde{A}$ and $\tilde{B}$ is the $L$-quantization of the respective row of $A$ and $B$. Lastly, define $\mathcal{P}_{m,L}$ to be the set of all quantized probability matrices derived from $\mathcal{P}_m$.*

Via counting arguments, the cardinality of $P_{m,L}$ is bounded by $|P_{m,L}| \leq (L+1)^{2km}$. This quantized family gives us the following approximation ability.

**Lemma 7** (De-quantization). *For a probability vector $p$, $L$-quantization satisfies $|p_i - \tilde{p}_i| \leq \frac{1}{L}$ for all $i$, and $\|p - \tilde{p}\|_1 \leq \frac{2}{L}$. For a conditional probability matrix $Q \in \mathcal{P}_m$, any quantization $\tilde{Q}$ satisfies $|Q_{ij} - \tilde{Q}_{ij}| \leq \frac{3}{L}$ for all $i$. Furthermore, if $Q > \epsilon$ per entry and $L > \frac{6}{\epsilon}$, then:*

$$|R(Q) - R(\tilde{Q})| \leq \frac{6}{L\epsilon} \quad \text{and} \quad |R_n(Q) - R_n(\tilde{Q})| \leq \frac{6}{L\epsilon}.$$

We now give a PAC-style relative uniform convergence bound on the empirical risk [27].

**Theorem 8** (Relative uniform convergence). *Assume lower-bounded $P > \delta$ and choose any $\tau > 0$. We then have the following uniform bound over all lower-bounded $\tilde{Q} > \epsilon$ in $\mathcal{P}_{m,L}$ (Definition 6):*

$$\Pr\left\{ \sup_{\tilde{Q} \in \mathcal{P}_{m,L}, \tilde{Q} > \epsilon} \frac{R(\tilde{Q}) - R_n(\tilde{Q})}{\sqrt{R(\tilde{Q})}} > \tau \right\} \leq e^{-\frac{n\tau^2}{20\log\frac{1}{\epsilon} + 2\tau\sqrt{10\frac{1}{\delta}\log\frac{1}{\epsilon}}} + 2km\log(L+1)}. \tag{4}$$

The proof of this Theorem consists, for fixed $\tilde{Q}$, of showing a sub-Poisson concentration of the sum of the log likelihood ratios. This needs care, as a simple Bennett or Bernstein inequality is not enough, because we need to eventually self-normalize. A critical component is to relate the variance and scale of the concentration to the KL-risk and its square root, respectively. The theorem then follows from uniformly bounding the normalized variance and scale over $\mathcal{P}_{m,L}$ and a union bound.

To put the pieces together, first note that thanks to the fact that the optimum is also a stable point of the ADD-$\frac{1}{2}$-SMOOTHED LOW-RANK, the add-$\frac{1}{2}$ nature of the updates implies that all of the elements of $Q_n$ are lower-bounded by $\frac{1}{2n+k}$. By Lemma 7 and a proper choice of $L$ of the order of $(2n+k)^2$, the quantized version won't be much smaller. We can thus choose $\epsilon = \frac{1}{2n+k}$ in Theorem 8 and use our assumption of $\delta = \frac{km}{n} \log(2n+k)$. Using Lemmas 5 and 7 to bound the contribution of the empirical risk, we can then integrate the probability bound of (4) similarly to the realizable case. This gives a bound on the expected risk of the quantized version of $Q_n$ of order $\frac{km}{n} \log \frac{1}{\epsilon} \log L$ or effectively $\frac{km}{n} \log^2(2n+k)$. We then complete the proof by de-quantizing using Lemma 7.

# 7 Experiments

Having expounded the theoretical merit of properly smoothing structered conditional probability matrices, we give a brief empirical study of its practical impact. We use both synthetic and real data. The various methods compared are as follows:

- Add-$\frac{1}{2}$, directly on the bigram counts: $\hat{P}_{n,ij}^{\mathsf{Add}\text{-}\frac{1}{2}} = (C_{ij} + \frac{1}{2})/(C_i + \frac{1}{2})$
- Absolute-discounting, directly on the bigram counts: $\hat{P}_n^{\mathsf{AD}}(C, \alpha)$ (see Section 5)
- Naive Add-$\frac{1}{2}$ Low-Rank, smoothing the counts then factorizing: $Q^{\mathsf{Naive\ Add}\text{-}\frac{1}{2}\text{-}\mathsf{LR}} = Q^{\mathsf{LR}}(C + \frac{1}{2})$
- Naive Absolute-Discounting Low-Rank: $Q^{\mathsf{Naive\ AD}\text{-}\mathsf{LR}} = Q^{\mathsf{LR}}(n\hat{P}_n^{\mathsf{AD}}(C, \alpha))$
- Stupid backoff (SB) of Google, a very simple algorithm proposed in [9]
- Kneser-Ney (KN), a widely successful algorithm proposed in [17]
- Add-$\frac{1}{2}$-Smoothed Low-Rank, our proposed algorithm with provable guarantees: $Q^{\mathsf{Add}\text{-}\frac{1}{2}\text{-}\mathsf{LR}}$
- Absolute-Discounting-Smoothed Low-Rank, heuristic generalization of our algorithm: $Q^{\mathsf{AD}\text{-}\mathsf{LR}}$

The synthetic model is determined randomly. $\pi$ is uniformly sampled from the $k$-simplex. The matrix $P = AB$ is generated as follows. The rows of $A$ are uniformly sampled from the $k$-simplex. The rows of $B$ are generated in one of two ways: either sampled uniformly from the simplex or randomly permuted power law distributions, to imitate natural language. The discount parameter is then fixed to $0.75$. Figure 1(a) uses uniformly sampled rows of $B$, and shows that, despite attempting to harness the low-rank structure of $P$, not only does Naive Add-$\frac{1}{2}$ fall short, but it may even perform worse than Add-$\frac{1}{2}$, which is oblivious to structure. Add-$\frac{1}{2}$-Smoothed Low-Rank, on the other hand, reaps the benefits of both smoothing *and* structure.

Figure 1(b) expands this setting to compare against other methods. Both of the proposed algorithms have an edge on all other methods. Note that Kneser-Ney is not expected to perform well in this regime (rows of $B$ uniformly sampled), because uniformly sampled rows of $B$ do not behave like natural language. On the other hand, for power law rows, even if $k \gg n$, Kneser-Ney does well, and it is only superseded by Absolute-Discounting-Smoothed Low-Rank. The consistent good performance of Absolute-Discounting-Smoothed Low-Rank may be explained by the fact that absolute-discounting seems to enjoy some of the competitive-optimality of Good-Turing estimation, as recently demonstrated by [22]. This is why we chose to illustrate the flexibility of our framework by heuristically using absolute-discounting as the smoothing component.

Before moving on to experiments on real data, we give a short description of the data sets. All but the first one are readily available through the Python NLTK:

- tartuffe, a French text, train and test size: 9.3k words, vocabulary size: 2.8k words.
- genesis, English version, train and test size: 19k words, vocabulary size: 4.4k words
- brown, shortened Brown corpus, train and test size: 20k words, vocabulary size: 10.5k words

For natural language, using absolute-discounting is imperative, and we restrict ourselves to Absolute-Discounting-Smoothed Low-Rank. The results of the performance of various algorithms are listed in Table 1. For all these experiments, $m = 50$ and $200$ iterations were performed. Note that the proposed method has less cross-entropy per word across the board.

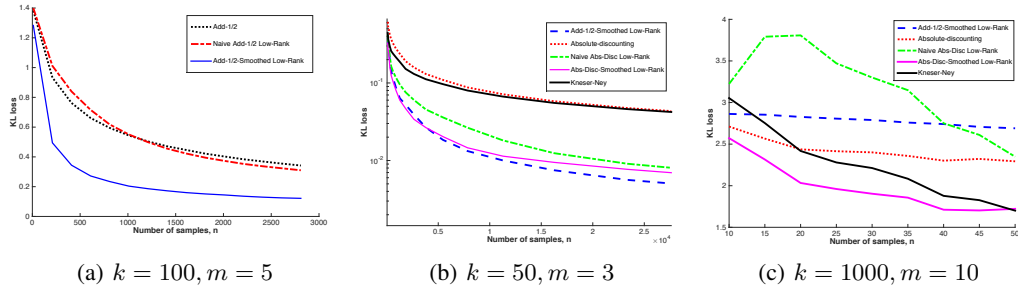

|     |     |     |
| :-: | :-: | :-: |
| (a) $k = 100, m = 5$ | (b) $k = 50, m = 3$ | (c) $k = 1000, m = 10$ |

Figure 1: Performance of selected algorithms over synthetic data

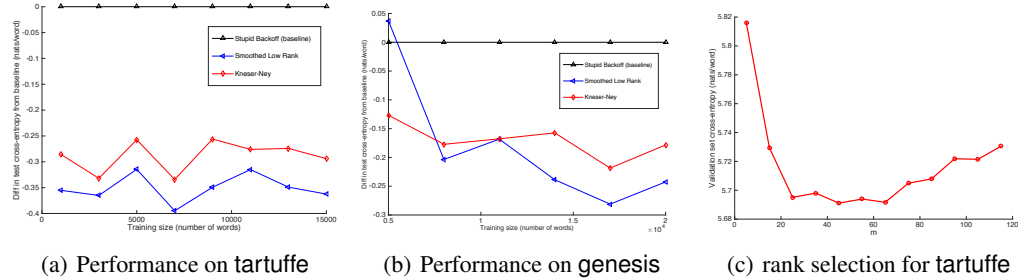

|     |     |     |
| :-: | :-: | :-: |
| (a) Performance on tartuffe | (b) Performance on genesis | (c) rank selection for tartuffe |

Figure 2: Experiments on real data

| Dataset | Add-$\frac{1}{2}$ | AD | SB | KN | AD-LR |
| --- | --- | --- | --- | --- | --- |
| tartuffe | 7.1808 | 6.268 | 6.0426 | 5.7555 | **5.6923** |
| genesis | 7.3039 | 6.041 | 5.9058 | 5.7341 | **5.6673** |
| brown | 8.847 | 7.9819 | 7.973 | 7.7001 | **7.609** |

Table 1: Cross-entropy results for different methods on several small corpora

We also illustrate the performance of different algorithms as the training size increases. Figure 2 shows the relative performance of selected algorithms with Stupid Backoff chosen as the baseline. As Figure 2(a) suggests, the amount of improvement in cross-entropy at $n = 15k$ is around 0.1 nats/word. This improvement is comparable, even more significant, than that reported in the celebrated work of Chen and Goodman [11] for Kneser-Ney over the best algorithms at the time.

Even though our algorithm is given the rank $m$ as a parameter, the internal dimension is not revealed, if ever known. Therefore, we could choose the best $m$ using model selection. Figure 2(c) shows one way of doing this, by using a simple cross-validation for the tartuffe data set. In particular, half of the data was held out as a validation set, and for a range of different choices for $m$, the model was trained and its cross-entropy on the validation set was calculated. The figure shows that there exists a good choice of $m \ll k$. A similar behavior is observed for all data sets. Most interestingly, the ratio of the best $m$ to the vocabulary size corpus is reminiscent of the choice of internal dimension in [20].

## 8 Conclusion

Despite the theoretical impetus of the paper, the resulting algorithms considerably improve over several benchmarks. There is more work ahead, however. Many possible theoretical refinements are in order, such as eliminating the logarithmic term in the sample complexity and dependence on $P$ (strong optimality). This framework naturally extends to tensors, such as for higher-order $N$-gram language models. It is also worth bringing back the Markov assumption and understanding how various mixing conditions influence the sample complexity. A more challenging extension, and one we suspect may be necessary to truly be competitive with RNNs/LSTMs, is to parallel this contribution in the context of generative models with long memory. The reason we hope to not only be competitive with, but in fact surpass, these models is that they do not use distributional properties of language, such as its quintessentially power-law nature. We expect smoothing methods such as absolute-discounting, which do account for this, to lead to considerable improvement.

**Acknowledgments** We would like to thank Venkatadheeraj Pichapati and Ananda Theertha Suresh for many helpful discussions. This work was supported in part by NSF grants 1065622 and 1564355.

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
