[Reviews · NeurIPS 2016]

Reviewer 1

Summary

In this paper, the authors considered an important problem in learning multiple related probability distribution. The authors adopted low-rank assumption to the probability matrix and use KL-divergence to measure the loss. The alternative minimization method with add-1/2-smoothing was proposed. The algorithm was proved to achieve near-optimal performance. Finally, the numerical experiments were provided to illustrate the power of the proposed procedure.

Qualitative Assessment

1. I think the authors should explain what is "PAC-style bound." If it is too long, giving some representative references will be helpful. 2. The results are of KL-divergence risk, how about other losses, such as Frobenius norm loss? Some discussions might be helpful. 3. In order to implement the proposed algorithm, one needs to specify the rank k. How to select the value k empirically? It would be better if the authors can provide some explanation here.

Confidence in this Review

2-Confident (read it all; understood it all reasonably well)


Reviewer 2

Summary

Given two discrete random variables X and Y (each taking k values), the paper considers the problem of estimating the k X k matrix P defined by P(i, j) = P(Y = j| X = i) from n i.i.d observations (X_s, Y_s), s = 1,..,n under the assumption that P can be factorized as AB for two similar stochastic matrices A and B of orders k X m and m X k respectively. The paper proposes an alternating optimization scheme for estimating P. A complication arises from the fact that many of the observed counts may be zero and the author propose at least two different ways of tackling it.

Qualitative Assessment

If my understanding is correct, Theorem 1 of the authors does not quite apply to their algorithm ADD-1/2-Smoothed Low-Rank. Instead, it applies to the non-computable algorithm where they assume that they have a minimizer of the objective function in Theorem 3. It is not clear if the alternating optimization algorithm proposed in the paper is guaranteed to converge to a minimizer of the objective in Theorem 3. If this is true, the authors should mention this before stating Theorem 1 to avoid misleading the reader. The "discounting" seems important from the Experiments section but this is not described in the main paper. If this is so important, the authors should make room for this in the main paper. The main results (Theorem 1 and 2) are not so surprising given that this is almost a parametric estimation problem with ~mk parameters (so the rates should be km/n). The interesting aspect here is if there would be log terms in the minimax rate. Theorem 1 has log-factors while Theorem 2 has none. It will be great if the authors can determine if there should be a log term or not in the minimax rate. It would also be helpful if the authors can comment on how to choose m somewhere in Section 5. It appears from their experiments that they do this via cross validation but a careful specification in Section 5 would be appreciated.

Confidence in this Review

2-Confident (read it all; understood it all reasonably well)


Reviewer 3

Summary

This paper studies the problem of learning low-rank probability matrices. The theoretical component provides sample upper bounds (that are nearly tight). The experimental component evaluates the performance of a known alternating minimization type algorithm.

Qualitative Assessment

This is a nice paper in an interesting topic. The contribution is above the threshold for NIPS, in my opinion. I feel that the experimental results are more interesting than the sample complexity bound (which is expected).

Confidence in this Review

2-Confident (read it all; understood it all reasonably well)


Reviewer 4

Summary

This paper studied the probability matrix in language bigram model. It assumed a low rank structure on the conditional probability matrix (P) and consider the estimation problem within the framework of non-negative matrix factorization. The author proposed a ADD-1/2-SMOOTHED LOW-RANK algorithm to estimate the probability matrix and justified its performance under the criterion of minimax KL distance. Both synthetic and real data studies were conducted to compare the practical performance of the proposed methods with several related methods.

Qualitative Assessment

Some of the statements are not very clear. The author should have provide more information. For example, what is the definition of polylog? It appears in line 60 as polylog(n) and later above line 116 as polylog(n,m,k). Besides, I wonder why the polylog(n,m,k) in Theorem 1 does not occur anywhere in the proof of Theorem 1. Another term needs to clarify is "PAC-style bound". The author could provide some background or add some reference to make it more clear. The structure of this paper is also a little confusing, as it shows the main result (section 4) of the ADD-1/2-SMOOTHED LOW-RANK algorithm before even introduce it (section 5). I suggest the author to explicitly give the proof of Theorem 2 or relative reference. Line 136 stating that the bound is based on additional information of the latent variable Z, while section 3 does not mention anything about this Z. So it is not very clear why 'the overview given in section 3 is sufficient' for the proof (line 173).

Confidence in this Review

2-Confident (read it all; understood it all reasonably well)


Reviewer 5

Summary

The paper incorporates add-1/2 and absolute smoothing, in to low-rank estimation of probability matrices, specifically for bigram word level models. The main contribution is theoretic with strong new results. Experiments on real and synthetic data sets show promising results.

Qualitative Assessment

Despite not fully understanding the theory, I think the paper is accessible and the addition of experiments makes it much nicer. Thus I recommend for acceptance.

Confidence in this Review

1-Less confident (might not have understood significant parts)


Reviewer 6

Summary

The paper proposes an algorithm for smoothing probability matrices with some underlying low rank sructure. The row rank paradigm is extremely common in many areas of Machine Learning (for example NLP) and this makes the proposed approach very useful. The paper provides a clear description of the ideas underlying the algorithm and a theoretical analysis of it (including sample complexity bounds). Some relevant NLP experiments are also provided.

Qualitative Assessment

Technical quality: the paper is very well supported both theoretically (through proof of relevant theorems) and experimentally (through the comparison to standard and widely used alternative smoothing approaches). Novelty/originality: The paper provides a novel contribution in a largely explored field. Potential impact or usefulness: Smoothing structured conditional distribution is a very common problem, so it is very likely that an efficient algorithm with provable guarantees to perform such task will attract significant interest. Clarity and presentation: Very good and often intuitive explanations make the paper a pleasent read.

Confidence in this Review

2-Confident (read it all; understood it all reasonably well)